# Soils rich in biological ice nucleating particles overproportionately abound in those resembling macromolecules produced by fungi

Franz Conen[1], Mikhail V. Yakutin[2]

[1]Department of Environmental Sciences, University of Basel, Bernoullistr. 30, 4056 Basel, Switzerland
[2]Institute of Soil Science and Agrochemistry, Siberian Branch of the Russian Academy of Sciences, Academician Lavrentyev Avenue, 8/2, 630090 Novosibirsk, Russia

*Correspondence to*: Franz Conen (franz.conen@unibas.ch)

**Abstract.** Soil organic matter carries ice nucleating particles (INP) of which the origin is hard to define and that are active at slight supercooling. The discovery and characterisation of INP produced by the widespread soil fungus *Mortierella alpina* permits a more targeted investigation of the likely origin of INP in soils. We searched for INP with characteristics similar to those reported for *M. alpina* in 20 soil samples from four areas in the northern midlatitudes and one area in the tropics. In the 15 samples where we could detect such INP, they constituted between 1 and 94% (median 11%) of all INP active at -10 °C or warmer (INP$_{-10}$) associated with soil particles < 5 µm. Their concentration increased overproportionately with the concentration of INP$_{-10}$ in soil and seems to be larger in colder climates. Large regional differences and prevalently high concentrations allow to make inferences regarding their potential role in atmosphere and soil.

## 1 Introduction

Soils could be a relevant source of ice nucleating particles (INP) found in the atmosphere and INP from soils are also found in precipitation (Creamean et al., 2013, 2014) and in rivers (Moffett, 2016; Larsen et al., 2017). Organic matter, or biological residues, associated with soil particles may contribute a major share to atmospheric INP active at temperatures warmer than -10 °C (Schnell and Vali, 1976; Szyrmer and Zawadzki, 1997; Conen et al., 2011, O'Sullivan et al., 2014; Creamean et al., 2013; Tobo et al., 2014, Hill et al., 2016). Recent progress in this field of research has been made by the detailed characterisation of INP produced by the widespread soil fungus *Mortierella alpina* (Fröhlich-Nowoisky et al., 2015). Plating and cultivation have allowed to identify *M. alpina* as the INP-producing organism through DNA sequencing followed by phylogenetic analysis. Together with the earlier discovery of *Fusarium avenaceum* and *Fusarium acuminatum* as sources of INP with similar characteristics (Pouleur et al., 1992), this new INP source raises the question of the more general relevance of cell-free INP produced by fungi in soils. Trying to identify and count, or determine the mass of these fungi in soil could be one approach. However, this approach would not account for the fact that INP produced by the organisms can be washed off, may be preserved, accumulate in the soil, and be exported from a watershed during intense rainfall (Larsen et al., 2017).

In a first attempt to gauge the potential relevance of cell-free, ice-nucleating macromolecules likely derived from fungi, we looked for INP in soils that match the challenge tests described for *M. alpina* (Fröhlich-Nowoisky et al., 2015).

## 2 Material and methods

We collected grab samples (100 g to 300 g of material per sample) from the surface of arable soils (Table 1) in Novosibirsk (Western Siberia), Saskatoon (Saskatchewan) and Colmar (France), from grasslands in La Brévine (Switzerland), and tropical mountain forests around Ranau (Borneo). Where present, aboveground vegetation and litter were removed before sampling. Samples were air dried and dry sieved (< 63 µm). From each sample 1 g of dry particles (< 63 µm) was weighed into a 50 ml centrifuge tube containing 20 ml of 0.1% NaCl, was shaken for 2 min by hand and allowed to settle for another 10 min. About 10 ml suspension were withdrawn from the top of the suspension and passed through a syringe filter with 5 µm pore size (sterile cellulose acetate filter; Sterlitech Corporation, Kent, USA), 9 ml of it into a pre-weighed aluminium tray, 1.0 to 1.5 ml into another tube together with the proper amount of 0.1% NaCl to create a 1:20 dilution of the suspension. The tray and its content were dried at 80 °C, re-weighed and the mass of particles < 5 µm determined from the difference to a control tray prepared with only NaCl solution. The tube containing the 1:20 dilution of the suspension was analysed for INP on a freezing nucleation apparatus (Stopelli et al., 2014) in 52 aliquots of 100 µl in 0.5 ml tubes and, if necessary, further diluted to a concentration at which most, but not all of the 52 tubes were frozen at -10 °C. Final concentrations of particles < 5 µm ranged from 0.02 to 15.5 µg ml$^{-1}$ with a median of 1.0 µg ml$^{-1}$. The remainder of the suspension with the final concentration was then passed through a 0.22 µm syringe filter (same material and supplier as above) and partitioned into three portions. One portion was analysed for INP without further treatment, the other two portions were either heated to 60 °C or 95 °C for 15 min in a water bath before being analysed the same way. From the original suspension and a 6 M solution of guanidinium chloride (> 99.5%; Roth GmbH + Co. KG, Karlsruhe, Germany) we prepared a similarly diluted suspension of particles < 0.22 µm and analysed it for INP after 1 to 2 hours of storage at room temperature. Guanidinium chloride deactivates bacterial and fungal INP, but not pollen (Pummer et al. 2012, 2015). Blank samples of 0.1% NaCl solution did not freeze at -10 °C. Our criteria for what we presume are cell-free fungal INP were an activation temperature of -6.5 °C or warmer (INP$_{-6.5}$) that is retained after heating to 60 °C, but which is deactivated by heating to 95 °C and by 6 M guanidinium chloride. For practical reasons (smallest mesh filter size available) we relaxed the size criterion (< 300 kDa) in Fröhlich-Nowoisky et al. (2015) to < 0.22 µm. This may seem generous, but still excludes other potential INP$_{-6.5}$ that are associated with cells and are not detached macromolecules. However, bacterial INP have been found to not withstand heating to 60 °C, with the exception of ice-nucleating entities resistant to boiling produced by *Lysinibacillus sp.* (Failor et al., 2017). Pollen derived INP macromolecules are not sensitive to guanidinium chloride or boiling (Pummer et al. 2012, 2015). Thus, there are currently no INP from other than fungal sources known to match the applied test criteria.

## 3. Results and discussion

We found what we presume are cell-free fungal INP in all samples with more than 1 INP active at -10 °C µg$^{-1}$ particles < 5 µm (INP$_{-10}$) (Fig. 1; Fig. 2). There might also have been a contribution of cell-free fungal INP in samples with less than 1 INP$_{-10}$ µg$^{-1}$, but it was too small to be detected. The latter applies to all 4 samples from tropical Ranau (INP$_{-10}$ < 0.1 µg$^{-1}$) and one (of 3) from the wine growing area around Colmar (INP$_{-10}$ = 0.3 µg$^{-1}$). Higher concentrations of INP in cold, compared to warm regions were previously reported by Schnell and Vali (1976) and Augustin et al. (2013). Guanidinium chloride reduced the number of INP$_{-10}$ in all suspensions of particles < 0.22 µm to below the detection limit (to 2% or less of what we found in suspensions prepared with 0.1% NaCl), so did heating to 95 °C. What we presume are cell-free fungal INP were therefore not derived from pollen (Pummer et al., 2012, 2015) or *Lysinibacillus sp*. (Failor et al., 2017), otherwise they would still have been active after heating to 95 °C. Averaged over all samples 97% (+/- 9%) of INP$_{-6.5}$ associated with particles < 5 µm passed through the 0.22 µm filter and 86% (+/-10%) of those remained active after heating to 60 °C. Consequently, about 5/6$^{th}$ (0.97 x 0.86 = 0.83) of all INP$_{-6.5}$ found in soil particles < 5 µm matched characteristics of cell-free fungal INP. There might have been a small contribution by *Isaria farinosa* (Huffman et al., 2013) to the number of INP$_{-6.5}$ determined before heat treatment. However, these INP would have been deactivated after heating to 60 °C (Pummer et al., 2015) and would not have contributed to the number of cell-free fungal INP considered in the present study. Smaller fractions of INP$_{-10}$ passed the challenge tests. On average 81% (+/- 6%) passed through 0.22 µm and only half (51%, +/- 9%) of all INP$_{-10}$ were also active after heating to 60 °C.

Cell-free fungal INP made up only 1/20$^{th}$ of INP$_{-10}$ around Colmar, but 2/3$^{rd}$ of INP$_{-10}$ around Novosibirsk. Regression analysis of the ensemble of 15 samples with detectable cell-free fungal INP from all four areas on three continents (Fig. 1) suggests that a doubling of INP$_{-10}$ may be associated with a tripling of the number of cell-free fungal INP ($2^{1.6}$ = 3). This trend not only applies across the different areas investigated, but also within certain areas (Saskatoon, La Brévine). We speculate that the great plasticity in the contribution of cell-free fungal INP results from their property of being macromolecules that can be washed-off from the mycelium. In principle, the production of a macromolecule requires less resources than the production a complete cell carrying an ice-nucleation active entity. Hence, an organism capable to release ice-nucleation active macromolecules has a greater range over which it can potentially modify its surrounding in terms of ice nucleation.

## 4. Inferences from a wider context

### 4.1. Atmosphere

The frequency of clouds containing ice particles at moderate supercooling is larger above fertile land than downwind a desert (Kanitz et al., 2011). One cause of this difference could be higher concentrations of INP in fertile soils compared to desert soils (Conen et al., 2011). Above fertile land, however, atmospheric INP concentrations do not seem to vary with INP

concentration in soils. The one to two orders of magnitude larger concentrations of INP in soils at Novosibirsk, compared to soils in La Brévine and Colmar (Fig. 1), leaves little or no trace in the atmosphere. At Chaumont, a site 30 km to the East of La Brévine and 120 km to the South-west of Colmar, we observed similar concentrations of atmospheric INP active at -8 °C or warmer as we did in Novosibirsk (Conen et al., 2017). During April and May, when arable soils are prepared for seeding

and wind erosion is most prevalent (Selegey et al., 2011), median values were 4 INP $m^{-3}$ and 7 INP $m^{-3}$ at Chaumont and Novosibirsk, respectively. Thus, soils are unlikely the dominant source of biological INP in the atmosphere above the cropland belt stretching from Western Europe eastward all the way to Novosibirsk. Still, the influence of soils as a source of atmospheric INP might appear unduly small in this comparison because of efficient atmospheric mixing within the latitudinal band. Nevertheless, it is likely that vegetation and leaves decaying at the soil surface make a larger contribution to

atmospheric $INP_{-10}$ (Schnell and Vali, 1976; Conen et al., 2017). We conclude that cell-free fungal INP associated with soil dust probably have a minor influence on ice formation in supercooled clouds and regional differences between soils are masked by atmospheric mixing and relatively larger contributions of INP from vegetation and decaying leaves.

## 4.2 Soil

Postulated potential advantages to an organism capable to catalyse ice formation at slight supercooling include the cleavage of structures by ice formation to access otherwise occluded resources (Paul and Ayres, 1991) and the accumulation of water through growing ice from vapour in the surrounding air (Kiefft, 1988). However, there is little experimental evidence to support these ideas in the context of soil. To our knowledge, the most convincing evidence for an accumulation of water in the form of ice was described by Hofmann et al. (2015). Fascinating sculptures of hair ice can form on the surface of dead

wood infected by the fungus *Epidiopsis effusa* through the mechanism of ice segregation. This mechanism transports slightly supercooled water from inside the wood to a body of ice growing on the outside of it. The heat released by the phase transition stabilises the front between liquid and ice, as long as water is supplied to the growing ice at a sufficient rate. Although fungal activity is responsible for shaping hair ice, ice segregation proceeds under the same conditions equally without the fungus, but then results in an ice crust. Temperatures recorded by Hofmann et al. (2015) inside and outside of

wood samples showed that hair ice formation started when temperatures had decreased to about -0.5 °C in one experiment, and to -2.5 °C in another experiment. In both cases temperature inside the wood increased sharply after the onset of ice formation and stabilised near -0.2 °C through the heat released by ice formation, while temperature outside the wood continued to decrease. In one of the experiments, ice growth stopped when outside temperature had decreased to -4 °C.

The same process of ice segregation as described by Hofmann et al. (2015) may also take place at the surface or within the porous structure of soil, where larger pores are typically air-filled and water is held in finer capillaries, similar to those supplying water to the hair ice growing on wood. Visible phenomena of water accumulating through ice segregation at or near the soil surface include ice needles and ice lenses (Dash et al., 2006). For a soil fungus to benefit from ice segregation, it has to produce INP active as close as possible to 0 °C. We think that $INP_{-6.5}$ do not provide much of an advantage in this

context. Even in a very small volume of soil, pore water is unlikely to supercool to that temperature. Further, the volume of water that might be harvested through ice segregation would be irrelevantly small, if there are other INP active at the same temperature nearby, which is definitively the case for all samples shown in Figure 1. It can only be the much rarer INP active closer to 0 °C that potentially provide the advantage of ice segregation to an INP-producing fungus in soil. Pouleur et al.

(1992) found about 1 in $10^4$ $INP_{-6.5}$ was already active at -2.5 °C. The large numbers of cell-free fungal INP found in our samples may just be a proxy for the soil-ecologically relevant INP active closer to 0 °C. The detection of latter would require larger volumes of soil (e.g. millimetre-size aggregates) tested under conditions where temperature can be controlled with great stability and high precision (e.g. within a dry-block temperature calibrator).


**Acknowledgements**

The collaboration between the authors on the issue of ice nuclei in soils was supported by the Swiss National Science Foundation through grant number IZK0Z2-142484/1 to MVY for a short visit to Basel in summer 2012, during which much of the method applied in this study was developed. We thank K. Blomquist, V. P. Baranov and S. Tresch for providing

samples from Saskatoon, Ranau and La Brévine, respectively. This manuscript has benefitted a lot from comments and suggestions made by Cindy Morris and a second, anonymous Referee.

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

**Table 1:  Details of sample origin, including mean annual temperature (MAT) and precipitation (MAP). In each area between 3 and 6 samples ($N$) were collected. A sample consisted of 100 to 300 g of soil collected from the surface within a radius of a few meters. The maximum distance between samples ($D_{max}$) ranged from 4 to 106 km.**


| Location | | Novosibirsk | Saskatoon | La Brévine | Colmar | Ranau |
|---|---|---|---|---|---|---|
| Region | | Southwestern | Northern | Jura | Upper Rhine | Borneo |
| | | Siberia | Great Plains | Mountains | Valley | |
| Coordinates | latitude | 54$^o$38' to 55$^o$18' N | 52$^o$04' to 52$^o$08' N | 46$^o$59' N | 48$^o$00' to 48$^o$05' N | 05$^o$59' to 06$^o$03' N |
| | longitude | 82$^o$44' to 84$^o$23' E | 106$^o$29' to 106$^o$37' W | 06$^o$36' E | 07$^o$19' to 07$^o$23' E | 116$^o$42' E |
| Altitude | (m a.s.l.) | 120 - 150 | 500 - 520 | 1050 | 200 | 450 - 690 |
| Landuse | | arable crops | arable crops | grassland | arable crops | mountain forest |
| MAT | ($^o$C) | 1.7 | 2.6 | 4.9 | 10.9 | 27 |
| MAP | (mm) | 448 | 354 | 1597 | 607 | 2880 |
| Sampling | | May, Jun 2013 | Oct 2014 | Sep 14 | Oct, Nov 2014 | Mar 2014 |
| $N$ | | 6 | 3 | 4 | 3 | 4 |
| $D_{max}$ | (km) | 106 | 12 | 4 | 10 | 8 |


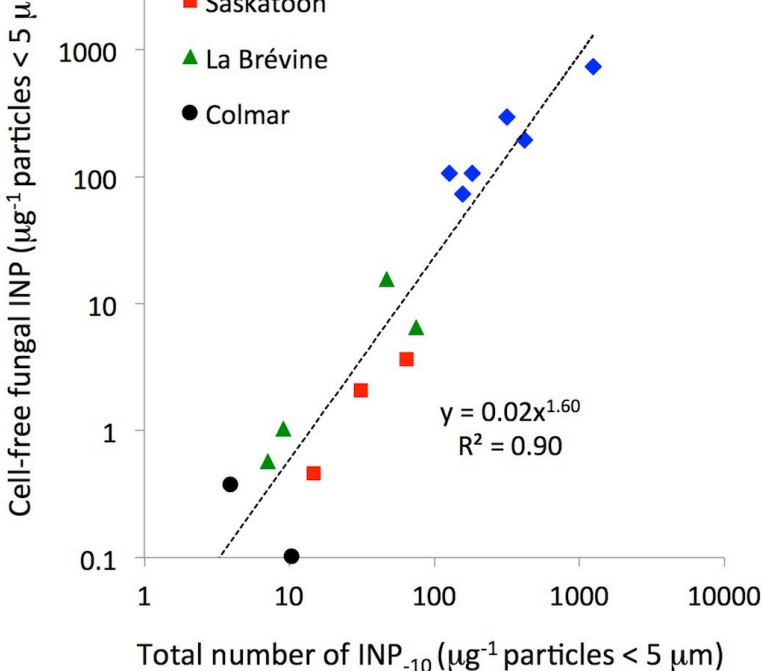




**Figure 1: Ice nucleating particles with characteristics of macromolecules released by certain fungi as a function of the total number of INP active at -10 °C in soil particles < 5 μm. The trendline was fitted to all data in the plot. Not plotted are 4 samples from tropical Ranau (INP$_{-10}$ < 0.1 μg$^{-1}$) and one (of 3) from the wine growing area around Colmar (INP$_{-10}$ = 0.3 μg$^{-1}$) in which we could not detect any INP resembling macromolecules released by fungi.**



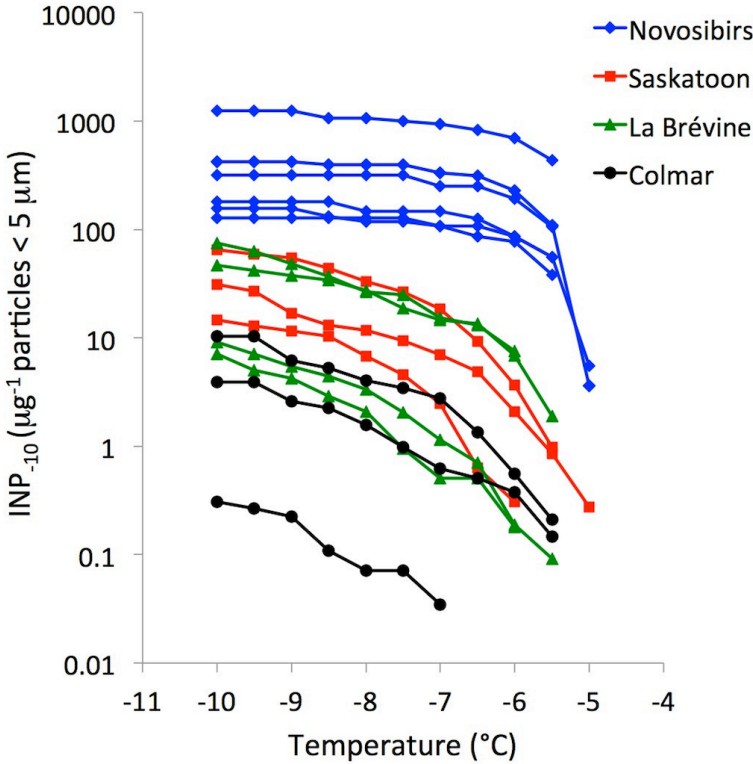

**Figure 2: Cumulative freezing spectra of the (untreated) samples shown in Figure 1 and a third sample from Colmar in which we**

 **could not detect INP resembling macromolecules released by fungi (lower most curve).**