# Peer review of "Soils rich in biological ice nucleating particles overproportionately abound in those resembling macromolecules produced by fungi"

_Biogeosciences, 2018_

## Referee Comment (RC1) · Anonymous Referee #1 · 27 Apr 2018

Review for "Potential relevance of Mortierella alpina as a source of ice nucleating particles in soil" by Franz Conen and Mikhail V. Yakutin, submitted to Biogeosciences Discuss.

This comparably short manuscript examines the ice nucleating ability of samples derived from six different soil samples collected in different parts of the world. The temperature range examined is restricted to comparably warm temperatures (down to -10°C), and different sample treatments are used to ascribe a fraction of the observed ice activity to "M-like" ice nucleators, i.e., to ice active entities which might be similar to those observed for the soil fungus Mortierella alpina. At the temperature of -10°C,

the fraction of these "M-like" ice active entities to the total number of ice active entities is then derived. It increases towards higher concentrations, much more than linearly, suggesting a possibly higher fraction of ice activity being contributed by "M-like" ice nucleators (possibly fungi) in colder areas.

The result that there are ice active entities in soils in different locations around the world is neither surprising nor totally new (the review paper by Szyrmer and Zawadzki (1997) already describes much older work on ice activity in soils). In recent times, this topic has been picked up again and refined, clearly identifying macromolecules related to soil fungi as ice active, and respective work is cited in the here presented manuscript. The new aspect of this new study is, that more locations are added where this ice activity in soils is found as well, together with the different treatments indicating that the observed ice nucleation may indeed be related to the abovementioned soil fungus. This is, however, only indirectly derived. The potentially higher ice activity of "M-like" ice nucleators in colder areas is also interesting but was described before (see my remarks below), albeit not to a great extent.

There are a few additions I would have wished for, and I give these below. However, I have to admit that I have difficulties in judging whether the content of the manuscript merits publication in Biogeosciences and would want to leave this final judgment for the editor, based on my remarks above.

Specific comments:

page 1, line 15: Literature on the statement that soils are a relevant source for ice nucleating particles (INP) would be good – while the presence of INP in soils is quite certain (see literature you cite in line 18), the transfer to the atmosphere is less well understood.

page 2, line 1: You call these INP "M. alpina like", and further down in the text it is justified some more why you assume that these are derived from fungi and not e.g., from bacteria, related to the treatments you do. It could help if you added here that the

assumption of a fungal origin is justified further down.

page 2, Methods: I would have liked to see a map which shows were the different samples were taken, at least roughly. Also: how much soil was originally sampled (e.g., in g of dried soil)? And in which depth (surface, further down, if it was the surface, was a plant cover removed, first)? This could be interesting information for people who would like to do similar experiments.

page 2, line 6: Why a NaCl solution and not pure water?

page 2, line 14: Put the units behind the numbers, i.e., "Final concentrations of particles < 5 $\mu$m ranged from 0.02 to 15.5 $\mu$g ml-1".

page 2, line 18: I wondered why you used guanidinium chloride, and this then only became clear at the end of this chapter. Please restructure the text so that the reader can know earlier what you are aiming at. The sentence on page 2, line 25-26 should be moved up.

page 2, line 29-31: Your findings here should be related to Schnell & Vali (1979), who report a dependency of the abundance of ice nuclei in leave litter on climatic zones, with higher abundancy in colder climates. A comparable relation has also been reported more recently for pollen in Augustin et al. (2013).

I would have liked to see some of the freezing curves. There are ample of them around in literature, but it is always good to look at them, as they tell stories, and as each study is different. This would be 15 curves (one each for each point shown in the present Fig. 1), and it would be nice to see if they are all similar, or different. Also, seeing the deactivation at least for selected samples and treatments presented as freezing curves would be good.

Literature:

Augustin, S., H. Wex, D. Niedermeier, B. Pummer, H. Grothe, S. Hartmann, L. Tomsche, T. Clauss, J. Voigtländer, K. Ignatius, and F. Stratmann (2013), Immersion

freezing of birch pollen washing water, Atmos. Chem. Phys., 13, 10989–11003, doi:10.5194/acp-13-10989-2013.

Schnell, R., and G. Vali (1976), Biogenic ice nuclei: Part I. Terrestrial and marine sources, J. Atmos. Sci., 33, 1554-1564.

Szyrmer, W., and I. Zawadzki (1997), Biogenic and anthropogenic sources of ice-forming nuclei: A review, BAMS, 78(2), 209-228.

---

## Referee Comment (RC2) · C. Morris (Referee) · 9 May 2018

GENERAL REMARKS

This concise manuscript cuts to the chase with regard to the question of what organisms contribute to the INPs active at temperatures above -10°C in the soil. The methods are based on the recently described chemical and heat sensitivities of the INPs produced by the fungus Mortierella alpina. The authors evaluate the part of the INPs of biological origin in various soils that correspond to these traits.

As the authors explain, INPs produced by M. alpina can be washed away from

mycelium and nevertheless maintain ice catalyzing activity. Therefore, the INPs could move around independently from the fungal tissues (spores or mycelium). This raises the possibility that soils could contain INPs produced by M. alpina without any other traces (e.g. DNA) of the fungus that could be used to validate co-occurrence of the fungus and its INPs.

The weakness of the approach used here is the lack of knowledge of the diversity of microorganisms that can produce INPs in soil. This is not the fault of the authors. Nevertheless, in light of the unexpected discoveries that can happen in this growing field, I think that it is short-sighted to assign this activity to any particular microorganism without some other type of validation.

I think that the authors could choose several strategies to valorize their results. On the one hand they could present this work as a sort of opinion-paper on how this question could be approached. In that case they should change the title to indicate that this is a manuscript about methods, they should change the name of the INPs that they are detecting (do not use the "M. alpina-like" subscript) and then add on discussion about the complementary experimental approaches that would provide additional corroborative data on the importance of M. alpina as the sources of these INPs. On the other hand, they could do additional experiments to provide these supplementary corroborative data and include them in a more comprehensive analysis of the underlying question. Other experimental approaches could include seeding soil with increasing quantities of M. alpina and testing for the presence of the fungus with DNA technologies in addition to characterizing the INPs in the soil. It would be strange for the fungus to be universally absent in soils where there are INPs with the traits the authors have targeted if this fungus is, indeed, at the origin of the INPs.

SPECIFIC REMARKS Please use italics when writing the latin name of the fungus.

p. 2, L 30 : Put "The" at the beginning of the sentence ("The latter. . . ")

p. 2, L 32: Add "the" to make the phrase "below the detection limit ".

---

## Author Comment (AC1) · 22 May 2018

We thank Referee #1 for assessing our manuscript and are grateful for all comments. The general remarks question the novelty of our study. We agree that earlier work has already convincingly shown the ubiquity of ice nucleating particles (INP) in soils around the world and merely adding more data from different soils does not necessarily justify a publication. However, our study clearly goes further. It shows that INP with characteristics similar to those of a recently described fungus over-proportionately contribute to INP concentrations in soils where they are generally higher than elsewhere. Admittedly, the discussion of this finding may have been too short. In addition to what

is already said in the manuscript, we speculate that the great plasticity in the contribution of cell-free fungal INP results from their property of being macromolecules that can be washed-off from the mycelium. In principle, the production of a macromolecule requires less resources than the production a complete cell carrying an INP. Hence, an organism capable to release such macromolecules has a greater range over which it can potentially modify its surrounding in terms of ice nucleation.

In response to comments by Cindy Morris (Referee), we made another effort to valorise the results in a wider context (please see our reply to Cindy Morris).

Replies to Specific comments

Thank you for the additional literature on soils as a relevant source of INP and earlier findings regarding the large abundance of INP in regions with a cold climate.

Showing the sampling locations on a map would unfortunately duplicate the much more precise information about the geographic locations shown in Table 1. We hope to give the reader a better idea of where the samples were taken by mentioning the regions to which the locations belong (i.e. Novosibirsk (Western Siberia), Saskatoon (Saskatchewan), Colmar (France), La Brévine (Switzerland), Ranau (Borneo).

Where present, aboveground parts of vegetation were removed and soil samples were collected with a small shovel from the surface of the soil. Each sample consisted of 100 g to 300 g of soil.

In our experiments we used a 0.1% NaCl solution instead of water because it improves the detection of phase change with our apparatus (Stopelli et al., 2014), especially at slight supercooling.

We added a Figure with the freezing curves (please see below). As for the deactivation, we can say that smaller fractions of INP active at -10 °C or warmer passed the challenge tests. On average 81% (+/- 6%) passed through 0.22 $\mu$m and only half (51%, +/- 9%) of all INP active at -10 °C or warmer were also active after heating to 60 °C.

Stopelli, E., Conen, F., Zimmermann, L., Alewell, C., and Morris, C. E.: Freezing nucleation apparatus puts new slant on study of biological ice nucleators in precipitation, Atmos. Meas. Tech., 7, 129-134, doi:10.5194/amt-7-129-2014, 2014.

———————————————————————

[Figure]

Figure: Freezing curves of untreated samples. Plot of INP$_{-10}$ ($\mu g^{-1}$ particles < 5 $\mu$m) versus Temperature (°C), with legend entries Novosibirsk, Saskatoon, La Brévine, Colmar.

**Fig. 1.** Freezing curves of untreated samples

---

## Author Comment (AC2) · 22 May 2018

We appreciate your judgement and suggestions, which help us to refrain from interpreting our data with undue bias towards a newly characterised source of INP in soils.

We like the idea of an opinion paper but also think that our study contains new experimental findings going beyond an opinion. The opinionated aspect of the current manuscript is the attribution of the finding to a specific organism. In a revised version we would broaden the current category "INP-M-like" to "cell-free fungal INP" and change the title to "Soils rich in biological ice nucleating particles overproportionately abound in those resembling macromolecules produced by fungi".

[Figure]

To valorize the results, we propose to add the following section at the end of the current manuscript:

4. Inferences from a wider context

4.1. Atmosphere

The frequency of clouds containing ice particles at moderate supercooling is larger above fertile land than downwind a desert (Kanitz et al., 2011). One cause of this difference could be higher concentrations of INP in fertile soils compared to desert soils (Conen et al., 2015). However, regional atmospheric INP concentrations above fertile land do not seem to vary with INP concentration in soils. The one to two orders of magnitude larger concentrations of INP in soils at Novosibirsk, compared to soils in La Brévine and Colmar (Fig. 1), leaves little or no trace in the atmosphere. At Chaumont, a site 30 km to the East of La Brévine and 120 km to the South-west of Colmar, we observed similar concentrations of atmospheric INP active at -8 °C or warmer as we did in Novosibirsk (Conen et al., 2017). During April and May, when arable soils are prepared and wind erosion is most prominent (Selegey et al., 2011), median values were 4 INP mˆ-3 and 7 INP mˆ-3 at Chaumont and Novosibirsk, respectively. Thus, soils are unlikely the dominant source of biological INP in the atmosphere above the cropland belt stretching from Western Europe eastward all the way to Novosibirsk. Still, the influence of soils as a source of atmospheric INP might appear unduely small in this comparison because of efficient atmospheric mixing within the latitudinal band. Nevertheless, it is likely that vegetation and leaves decaying at the soil surface make a larger contribution to atmospheric INP (-10 °C) (Schnell and Vali, 1976; Conen et al., 2017). We conclude that cell-free fungal INP associated with soil dust probably have a minor influence on ice formation in supercooled clouds and regional differences between soils are masked by atmospheric mixing with relatively larger contributions of INP from vegetation and decaying leaves.

4.2 Soil

Postulated potential advantages to an organism able to catalyse ice formation at slight supercooling include the cleavage of structures by ice formation to access otherwise occluded resources (Paul and Ayres, 1991) and the accumulation of water through growing ice from vapour in the surrounding air (Kiefft, 1988). However, there is little experimental evidence to support these ideas in the context of soil. To our knowledge, the most convincing evidence for an accumulation of water in the form of ice was described by Hofmann et al. (2015). Fascinating sculptures of hair ice can form on the surface of dead wood infected by the fungus Epidiopsis effusa through the mechanism of ice segregation. This mechanism transports slightly supercooled water from inside the wood to a body of ice growing on the outside of it. The heat released by the phase transition stabilises the front between liquid and ice, as long as water is supplied to the growing ice at a sufficient rate. Although fungal activity is responsible for shaping hair ice, ice segregation proceeds under the same conditions equally without the fungus, but then results in an ice crust. Temperatures recorded by Hofmann et al. (2015) inside and outside of wood samples showed that hair ice formation started when temperatures had decreased to about -0.5 °C in one experiment, and to -2.5 °C in another experiment. In both cases temperature inside the wood increased sharply after the onset of ice formation and stabilised near -0.2 °C through the heat released by ice formation, while it continued to decrease on the outside. In one of the experiments, ice growth stopped when outside temperature had decreased to -4 °C.

The same process of ice segregation as described by Hofmann et al. (2015) may also take place at the surface or within the porous structure of soil, where larger pores are typically air-filled and water is held in finer capillaries, similar to those supplying water to the hair ice growing on wood. Visible phenomena of water accumulating through ice segregation at or near the soil surface include ice needles and ice lenses (Dash et al., 2006). For a soil fungus to benefit from ice segregation, it has to produce INP active as close as possible to 0 °C. We think that INP (-6.5 °C) do not provide much of an advantage in this context. Even in a very small volume of soil, pore water is unlikely to supercool to that temperature. Further, the volume of water that might be harvested

through ice segregation would be irrelevantly small, if there are other INP active at the same temperature nearby, which is definitively the case for all samples shown in Figure 1 (of our current manuscript). It can only be the much rarer INP active closer to 0 °C that potentially provide the advantage of ice segregation to an INP-producing fungus in soil. Pouleur et al. (1992) found about 1 in 10ˆ4 INP (-6.5 °C) was already active at -2.5 °C. The large numbers of cell-free fungal INP found in our samples may just be a proxy for the soil-ecologically relevant INP active closer to 0 °C. The detection of latter would require larger volumes of soil (e.g. millimetre-size aggregates) tested under conditions where temperature can be controlled with great stability and high precision (e.g. within a dry-block temperature calibrator).

References

Conen, F., Rodriguez, S., HuÌĹglin, C., Henne, S., Herrmann, E., Bukowiecki, N., and Alewell, C.: Atmospheric ice nuclei at the high-altitude observatory Jungfraujoch, Switzerland, Tellus B, 67, 25014, doi.org/10.3402/tellusb.v67.25014, 2015.

Conen, F., Yakutin, M. V., Yttri, K. E., and Hüglin, C.: Ice nucleating particle concentrations increase when leaves fall in autumn, Atmosphere 8, 202, doi:10.3390/atmos8100202, 2017.

Dash, J. G., Rempel, A. W., and Wettlaufer, J. S.: The physics of pre-melted ice and its geophysical consequences, Rev. Mod. Phys., 78, 695-741, doi:10.1103/RevModPhys.78.695, 2006.

Hofmann, D., Preuss, G., and Mätzler, C.: Evidence for biological shaping of hair ice. Biogeosciences 12, 4261-4273, doi:10.5194/bg-12-4261-2015, 2015.

Kanitz, T., Seifert, P., Ansman, A., Engelmann, R., Althausen, D., Casiccia, C., and Rohwer, E.G.: Contrasting the impact of aerosol at northern and southern midlatitudes on heterogeneous ice formation, Geophys. Res. Lett. 38, L17802, doi:10.1029/2011GL048532, 2011.

Kiefft, T. L.: Ice nucleation activity in lichens, Appl. Environ. Microbiol. 54, 1678-1681, 1988.

Paul, N. D., and Ayres, P. G.: Changes in tissue freezing in Senecio vulgaris infected by rust (Puccinia lagenophorae), Ann. Bot. 68, 129-133, 1991.

Pouleur, S., Richard, C., Martin, J.-G., and Antoun, H.: Ice nucleation activity in Fusarium acuminatum and Fusarium avenaceum, Appl. Environ. Microbiol. 58, 2960-2964, 1992.

Schnell, R., and Vali, G.: Biogenic ice nuclei: Part I. Terrestrial and marine sources, J. Atmos. Sci., 33, 1554-1564, 1976.

Selegey, T. S., Koutsenogii, K. P., Filonenko, N. N., Popova, S. A., and Lenkovskaya, T. N.: Space-time variability of the characteristics of aerosol in the city-suburbs system (for Novosibirsk as example), Chemistry for Sustainable Development 19, 289-293, 2011.

---

## Referee Comment (RC3) · C. Morris (Referee) · 28 May 2018

Bravo for the suggested improvements. I still have the lingering question about why the INPs should be attributed to fungi. Can you add justification for that?

---

## Author Comment (AC3) · 29 May 2018

The attribution of INP to fungi is based on a combined set of criteria, which is not matched by INP from any other source we are currently aware of. These criteria are a size < 0.22 micron, ice-nucleation activity at -6.5 °C or warmer, tolerant to heating to 60 °C, and deactivation by heating to 95 °C and by 6 M guanidinium chloride. Bacterial INP have been found to not withstand heating to 60 °C (Pummer et al., 2015) with the exception of ice-nucleating entities produced by Lysinibacillus sp. (Failor et al., 2017). However, unlike the INP we presume are derived from fungi, INP from Lysinibacillus sp. also withstand boiling (Failor et al., 2017). Pollen-derived INP are insensitive to

boiling or be 6 M guanidinium chloride (Pummer et al., 2012).

**References**

Failor, K. C., Schmale III, D. G., Vinatzer, B. A., and Monteil, C. L.: Ice nucleation active bacteria in precipitation are genetically diverse and nucleate ice by employing different mechanisms, ISME J., 11, 2740-2753, doi:10.1038/ismej.2017.124, 2017.

Pummer, B. G., Bauer, H., Bernardi, J., Bleicher, S., and Grothe, H.: Suspendable macromolecules are responsible for ice nucleation activity of birch and conifer pollen, Atmos. Chem. Phys., 12, 2541-2550, doi:10.5194/acp-12-2541-2012, 2012.

Pummer, B. G., Bundke, C., Augustin-Bauditz, S., Niedermeier, D., Felgitsch, L., Kampf, C. J., Huber, R. G., Liedl, K. R., Loerting, T., Moschen, T., Schauperl, M., Tollinger, M., Morris, C. E., Wex, H., Grothe, H., Pöschl, U., Koop, T., and Fröhlich-Nowoisky, J.: Ice nucleation by water-soluble macromolecules, Atmos. Chem. Phys.., 15, 4077-4091, doi:10.5194/acp-15-4077-2015, 2015.

---

## Author Response (AR2)

Dear Editor

Thank you for your positive decision and your helpful suggestions, which we have happily accepted to further improve clarity and readability of the manuscript.

Best regards,

Franz Conen